# Hop Extract Anti-Inflammatory Effect on Human Chondrocytes Is Potentiated When Encapsulated in Rapeseed Lecithin Nanoliposomes

**DOI:** 10.3390/ijms232012423

**Published:** 2022-10-17

**Authors:** Émilie Velot, Florent Ducrocq, Loïc Girardeau, Alain Hehn, Séverine Piutti, Cyril Kahn, Michel Linder, Arnaud Bianchi, Elmira Arab-Tehrany

**Affiliations:** 1IMoPA (Molecular Engineering and Articular Physiopathology), CNRS (French National Centre for Scientific Research), Université de Lorraine, F-54000 Nancy, France; 2LAE (Laboratoire Agronomie et Environnement), INRAE (French National Research Institute for Agriculture, Food and Environment), Université de Lorraine, F-54000 Nancy, France; 3LIBio (Laboratoire d’Ingénierie des Biomolécules), Université de Lorraine, F-54000 Nancy, France

**Keywords:** chondrocytes, inflammation, interleukin-1β, osteoarthritis, liposomes, drug delivery system, hop extract, bitter acids

## Abstract

Hop (*Humulus lupulus* L.) is a plant used as an ingredient in beer or employed for its anti-inflammatory properties. The cultivation of hops is currently dedicated to the brewing industry, where mainly female flowers are used, whereas aerial parts, such as leaves, are considered coproducts. Osteoarthritis is the most common musculoskeletal disease associated with low-grade cartilage inflammation. Liposomes have been shown to be promising systems for drug delivery to cartilage cells, called chondrocytes. The aim of our work was to vectorize hop extract valorized from coproducts as a therapeutic agent to alleviate inflammation in human chondrocytes in vitro. Liquid chromatography allowed the identification of oxidized bitter acids in a methanolic extract obtained from the leaves of Cascade hops. The extract was encapsulated in rapeseed lecithin nanoliposomes, and the physicochemical properties of empty or loaded nanoliposomes exhibited no difference. Increasing concentrations of the hop extract alone, empty nanoliposomes, and loaded nanoliposomes were tested on human chondrocytes to assess biocompatibility. The appropriate conditions were applied to chondrocytes stimulated with interleukin-1β to evaluate their effect on inflammation. The results reveal that encapsulation potentiates the hop extract anti-inflammatory effect and that it might be able to improve joint inflammation in osteoarthritis. Furthermore, these results also show that a “zero waste” chain is something that can be achieved in hop cultivation.

## 1. Introduction

Articular cartilage is a specialized connective tissue covering the surfaces of bones in diarthrodial joints. This hyaline cartilage is both avascular and aneuronal. It is made of cells called chondrocytes that synthetize their own surrounding extracellular matrix (ECM) [1,2]. The mechanical properties of articular cartilage depend on its ECM and allow joint surfaces to resist wear [1,3,4].

Osteoarthritis (OA) is associated with cartilage degeneration. It is the most common musculoskeletal disease, affecting more than 300 million patients worldwide [5,6]. Although OA mechanisms are still not fully understood, low-grade joint inflammation has been reported to participate in this progressive disease [7,8,9]. Pro-inflammatory cytokines, such as interleukin (IL)-1β, promote hyaline ECM degradation and prevent its synthesis. This catabolic process alters the articular chondrocyte phenotype and prevents the cells from restoring the ECM balance [10]. Cyclooxygenase (COX)-2, microsomal prostaglandin E_2_ synthase (mPGES)-1, and inducible nitric oxide synthase (iNOS) are among the mediators of inflammation induced by IL-1β [11,12]. COX-2 and mPGES-1 are both responsible for prostaglandin E_2_ (PGE_2_) synthesis, while iNOS produces nitric oxide (NO). PGE2 and NO also take part in ECM catabolism, leading to cartilage degradation [13,14].

As there is still no treatment that might regenerate cartilage, the present therapeutic strategy for managing OA is to treat its symptoms (joint pain and reduced joint functional capacities) to maintain the patient’s quality of life [5,6]. Indeed, articular chondrocytes have a very low regenerative capability and cannot return hyaline cartilage to its initial state after injury or degeneration [15,16]. To improve damaged cartilage, joint regenerative medicine strategies using cell-free therapy approaches have emerged to target chondrocytes [17]. The aim is to deliver therapeutic agents, such as growth factors, to chondrocytes to reinstate their articular phenotype. However, numerous active biomolecules cannot cross the ECM and reach chondrocytes due to their short half-life and their prompt removal from the joint space [18]. Active biomolecules can be preserved and delivered to target cells by using encapsulation systems, such as nanoliposomes [19].

Rapeseed lecithin obtained from agro-resources can be formulated to make nanoliposomes that are biocompatible systems for chondrocytes [19]. They have recently shown the promising potential to generate the sustained release and delivery of transforming growth factor (TGF)-β1, a growth factor able to maintain the chondrocyte articular phenotype in vitro [20]. Therefore, rapeseed lecithin nanoliposomes are delivery systems that can carry anti-inflammatory molecules to prevent joint inflammation [21,22,23]. In addition, rapeseed lecithin nanoliposomes are rich in essential fatty acids, including linolenic acids (ω-3), the intake of which alleviates inflammation [24,25,26].

Hop plants (*Humulus lupulus* L.) are cultivated for the brewing industry, where only female cones are considered valuable products due to the presence of bitter acids [27]. In the end, two-thirds of the plant (mainly leaves and stems) is unexploited, whereas the whole plant is known to contain a wide range of active biomolecules, such as anti-inflammatory compounds [28,29]. The valorization of other hop plant parts, such as leaves, might represent a real opportunity to improve OA [28,30], since hop compounds have already shown their efficacy in reducing inflammation in other pathologies [31,32,33,34]. Furthermore, the valorization of hop leaves, considered an agricultural by-product and currently discarded as organic waste, might serve territorial bioeconomic purposes.

The aim of our work was to formulate a batch of hop extract valorized from leaves as a therapeutic agent to alleviate inflammation in human chondrocytes in vitro. In this study, we focused on methanol-soluble molecules extracted from hop leaves, which are very interesting coproducts regarding their phytochemical contents. After content analysis, our hop extract was formulated to be used alone or encapsulated in rapeseed lecithin nanoliposomes. The physicochemical properties of loaded and unloaded nanoliposomes were also evaluated. Increasing concentrations of the hop extract alone, empty nanoliposomes, and nanoliposomes encapsulating the hop extract were tested on human chondrocytes in vitro to validate every condition of biocompatibility. Then, chondrocytes stimulated with IL-1β were cultured in the various culture conditions at suitable concentrations to evaluate their effect on inflammation. The culture conditions were biocompatible with chondrocytes, except for the highest concentrations of empty and hop-extract-loaded nanoliposomes. Hop extract and empty rapeseed lecithin nanoliposomes significantly decreased mPGES-1 mRNA expression, while only empty nanoliposomes had the same effect on COX-2 and iNOS mRNA expression. Remarkably, hop extract encapsulated in nanoliposomes showed an accentuated decrease in the three transcripts. The same trends as those observed for *mpges-1* were obtained after the measurement of PGE_2_ and NO concentrations. Our results show that the vectorization of our hop extract with rapeseed lecithin nanoliposomes potentiates its anti-inflammatory effect and that this system could become a valuable tool to improve joint inflammation in OA.

## 2. Results

### 2.1. Characterization of Methanol-Soluble Molecules from Hop Extract

The extraction of hop from leaves was performed using a methanol-based extraction method dedicated to isolating soluble molecules [35]. The isolated compounds from the obtained batch of hop extract were studied by ultra-high-performance liquid chromatography–mass spectrometry (UHPLC-MS).

UHPLC-MS analysis in negative mode showed six groups of molecules, of which three could be identified (Figure 1, peaks P1 to P6). Surprisingly, α and β bitter acids could not be observed in the extract. However, oxidized β acids could be detected in the extract, such as cohulupone and hulupone + adhulupone (P4 and P5, respectively, Figure 1). The third identified group of molecules included feruloyl quinic acid (P1, Figure 1). The three remaining major groups of molecules were not identified (P2, P3, and P6, Figure 1) by Compound Discoverer^TM^ software (version 3.3; Thermo Fisher Scientific, llkirch-Graffenstaden, France).

The extracted batch made from hop leaves was formulated to be weighed and stored for further experiments. It was used alone or encapsulated in rapeseed lecithin liposomes.

### 2.2. Physicochemical Characterization of Empty and Hop-Loaded Rapeseed lecithin Liposomes

Liposomes made of rapeseed lecithin obtained from agro-resources were used in an empty state (i.e., without loading) or were loaded with the hop extract. The physicochemical characterization of empty and hop-loaded liposomes was performed by dynamic light scattering (DLS) and transmission electron microscopy (TEM) (Figure 2).

The DLS technique allows the acquisition of the following physicochemical parameters for liposomes: average hydrodynamic particle diameter, size distribution, polydispersity index (PDI), and ζ-potential. Empty and hop-loaded liposomes presented comparable results. The average size (close to 120 nm) confirmed that these particles were nanoliposomes. The size distributions were homogeneous and narrow, as shown by the corresponding graphical representations and PDI values of less than 0.3. Moreover, the large negative ζ-potential values were excluded from the interval from −30 mV to +30 mV, meaning that both colloids were stable (Figure 2A).

The TEM technique allows the visualization of liposome size and morphology. As shown for DLS, empty and hop-loaded particles presented equivalent results. Liposomes displayed a nanometric size close to 100 nm with a quite spherical shape. The lipid bilayer was observable, indicating the vesicular nature of the particles and confirming that they were liposomes (Figure 2B).

### 2.3. Biocompatibility of Hop Extract and Empty and Hop-Loaded Rapeseed Lecithin Nanoliposomes

Basal lactate dehydrogenase (LDH) release from Ctrl cells allowed us to gauge the possible cytotoxicity of the other cell culture conditions. No significant cytotoxicity was observed for any of the hop extract concentrations used (Figure 3A), whereas the two highest empty nanoliposome concentrations were cytotoxic, with the LDH release increased by ~25% for NL 500 and ~35% for NL 1000 (Figure 3B). The highest NL + H concentration was also found to be cytotoxic, with an LDH increase of about one-third (Figure 3C).

Human chondrocyte metabolic activity was investigated by determining the capacity of living cell mitochondria to reduce tetrazolium salts of MTT (3-[4,5-dimethylthiazol-2-yl]-2,5-diphenyltetrazolium bromide) into formazan crystals. Cell proliferation was examined by measuring the DNA concentration, where an increase indicates cell growth. While the hop extract alone did not seem to influence chondrocyte metabolic activity (Figure 4A), it significantly increased cell proliferation for H 150 (~40%) and H 300 (~60%) (Figure 5A). As NL 500 and NL 1000 were shown to be cytotoxic (Figure 3B), they had a severe impact on chondrocyte metabolic activity and proliferation (Figure 4A and Figure 5B, respectively). Metabolic activity was significantly reduced by ~45% for NL 500 and ~70% for NL 1000 (Figure 4B). It was the same trend for cell proliferation, with a decrease of ~20% for NL 500 and ~57% for NL 1000 (Figure 5B). Although only the NL 1000 + H 300 condition was cytotoxic (Figure 3C), a substantial reduction in metabolic activity and proliferation was observed for this condition and also for NL 500 + H 150 (Figure 4C and Figure 5C, respectively), with an emphasized effect compared to empty nanoliposomes at the same lecithin concentration (Figure 4B and Figure 5B, respectively). The decreases in metabolic activity and proliferation were, respectively, ~70% for NL 500 + H 150 and ~85% for NL 1000 + H 300 (Figure 4C) and ~64% for NL 500 + H 150 and ~86% for NL 1000 + H 300 (Figure 5C).

The evaluation of cytotoxicity, metabolic activity, and proliferation demonstrated the harmlessness of the hop extract when used alone for human chondrocytes in vitro and at the selected concentrations. On the contrary, culture conditions with high rapeseed lecithin concentrations (500 and 1000 µg/mL) in empty nanoliposomes or nanoliposomes encapsulating hop extract affected cell viability by reducing metabolic activity and proliferation and by increasing LDH release. To continue our work, the next culture conditions were chosen for being innocuous to chondrocytes and containing the highest amounts of materials (hop extract and/or lecithin).

### 2.4. Effect of Hop Extract and Empty and Hop-Loaded Rapeseed Lecithin Nanoliposomes on Inflammation

As shown in the previous section, the cell culture conditions with equivalent concentrations that are biocompatible with human chondrocytes in vitro and that have the highest quantities of the molecules of interest are H 75, NL 250, and NL 250 + H 75. Consequently, these three conditions were selected in the following experiments to monitor their impact on chondrocyte inflammation.

Human chondrocytes were stimulated in vitro with 1 ng/mL IL-1β to induce the inflammatory process, such as the one found in OA [36]. Stimulated cells were also exposed to H 75, NL 250, and NL 250 + H 75 conditions for 6, 24, and 48 h to measure the expression of various inflammation markers. Unstimulated cells treated or not with the selected conditions allowed us to ensure that the inflammatory reaction had not been initiated by improper stimulation (Figure 6 and Figure 7).

The expression of the inflammation markers COX-2 and iNOS, whose transcription is early [37], was first evaluated after 6 h of cell culture (Figure 6A,B). Then, the expression of mPGES-1 [38], a later marker, was studied after 24 h (Figure 6C). Regardless of the analyzed transcript, IL-1β stimulation alone triggered strong inflammation marker expression compared to the Ctrl condition, with an increase of over 13 times for COX-2 and over 6 times for iNOS and mPGES-1 (Figure 6). When it was used alone on stimulated chondrocytes, hop extract showed no impact on early marker expression (Figure 6A,B), while it decreased mPGES-1 transcription by ~23% (Figure 6C). Diminished expression, close to 25%, was found with empty nanoliposomes for both COX-2 and iNOS (Figure 6A,B). Surprisingly, the NL 250 effect on mPGES-1 expression was close to that of H 75 (~20%) (Figure 6C). Nanoliposomes encapsulating hop extract had a significantly greater effect on all markers, with a decrease in expression of ~51% for COX-2 (Figure 6A), ~43% for iNOS (Figure 6B), and ~60% for mPGES-1 (Figure 6C).

PGE_2_ is an end product of the COX-2 and mPGES-1 pathway, as NO is for iNOS [39,40]. These products were quantified after 48 h of cell culture. IL-1β stimulation alone triggered the massive release of PGE_2_ (nearly 400 times) and NO (more than 6 times) compared to the Ctrl condition (Figure 7). When used alone on stimulated chondrocytes, empty nanoliposomes decreased PGE2 and iNOS by ~14% and ~20%, respectively (Figure 7A,B). Although the H75 condition had no effect on the COX-2 transcript (Figure 6A), it lowered mPGES-1 expression (Figure 6C), which seems to have an influence on PGE_2_, whose release was close to 21% (Figure 7A). Similarly, although the hop extract alone had no effect on the iNOS transcript (Figure 6B), it appears to be involved in the NO pathway by decreasing its production (~27%) (Figure 7B). The NL 250 + H 75 condition showed the most relevant results by diminishing PGE2 by ~39% and NO by ~60% (Figure 7A,B).

Taken together, these results show that empty nanoliposomes had a better effect than the hop extract alone on decreasing inflammation marker transcription; however, this seems to be reversed when it comes to the synthesis of inflammation end products. Regardless of the culture conditions, nanoliposomes encapsulating hop extract generated the most striking results by alleviating inflammation by up to 60%, whether for transcripts or end products.

## 3. Discussion

Nowadays, hop is mainly cultivated to harvest female cones, which are used as ingredients in beer brewing. However, this plant has also been well-known for its medicinal properties for thousands of years. It has been shown that several active molecules detected in hop may be considered for their antioxidant and/or antimicrobial activities. Among these compounds, prenylated polyphenols were shown to have many biological activities that might be useful for human health [27,28,31,41].

The biosynthesis of prenylated compounds depends on the genotype, the organs, the developmental stage, and the environmental conditions [42,43]. Genotypes improved for the brewing industry, such as the Cascade variety used in our project, contain only low levels of xanthohumol and are supposed to produce higher amounts of α and β acids [44,45,46]. These molecules were found to be synthesized in specialized structures called lupulin glands located on the female cone. This is currently the reason why these organs are mainly harvested. Several studies also showed that the same set of compounds can be synthesized in leaf surface glandular structures called trichomes [45,47]. It was reported that the concentration of the molecules can be variable and can be higher in the leaves than in the cones [42] or much lower [48]. In our study, the analysis of the extract realized from hop leaves evidenced the presence of prenylated compounds corresponding to oxidized bitter acids, but we could not detect xanthohumol in our conditions. This latter result is not consistent with previous studies dedicated to *Humulus lupulus* L. cv. Cascade. Indeed, Sarraf and collaborators found a small amount of xanthohumol in hop leaves cv. Cascade grown in Québec [43]. It is possible that our conditions (soil, greenhouse conditions, and developmental stage) did not allow xanthohumol production or that its concentration was too low to be detected. Nevertheless, the detected prenylated molecules, even if oxidized, exert anti-inflammatory activity when they are vectorized in nanoliposomes. A deeper analysis of the molecules present in the extract may help to identify the molecules responsible for this activity. However, it is also possible that the mix of these molecules leads to a synergistic effect and that other molecules present in lower concentration (below the detection level of our analytical methods) contribute to this effect.

To date, only a few studies have reported attractive and encouraging results of hop extract treatments to reduce joint inflammation in strictly inflammatory models, such as collagen- or zymosan-induced arthritis in mice [34,49]. Data concerning OA, a joint disease with low-grade inflammation, are scarcer. Hence, Stracke and collaborators showed that hop compounds prevented ECM degradation in an in vitro OA model of bovine chondrocytes [30]. The lack of studies regarding hop effects on joint diseases is rather baffling because this plant seems to be of interest for improving a broad range of non-articular pathologies associated with inflammation. Hop extracts are involved in immunomodulation to prevent systemic inflammation through an anti-inflammatory effect on peripheral blood mononuclear cells [32], macrophages [29,33,50], and dendritic cells [51]. They are able to decrease the brain inflammation associated with neurodegenerative diseases such as Alzheimer’s disease [52] or Parkinson’s disease [53]. In addition to inflammatory disorders, hop-derived bitter acids are efficient against metabolic disorders, which makes them potent therapeutic candidates to treat metabolic syndrome, diabetes, or cardiovascular diseases. Among their other medicinal properties, they also have the potential to inhibit cancer [31,54].

In this study, we first formulated a batch of hop extract valorized from leaves to test its biocompatibility at increasing concentrations (3 to 300 µg/mL) in human chondrocytes in vitro. When it was used alone, our methanolic hop extract was found to be harmless since it showed no cytotoxicity and no modification of cell metabolic activity at any concentration. The two highest concentrations (150 and 300 µg/mL) even increased cell growth but were excluded because the activation of chondrocyte proliferation is undesirable and can cause a loss of phenotype in healthy joint cartilage [55]. Then, we evaluated the hop extract’s ability (75 µg/mL) to alleviate inflammation in an IL-1β-induced OA model of the same cells. Although it had no impact on the expression of the early inflammation markers COX-2 and iNOS, it decreased the late marker mPGES-1 as well as the release of PGE_2_ and NO, which are inflammation end products. Thus, the examined hop extract had an anti-inflammatory effect on human chondrocytes in vitro.

Our methanolic hop extract contained cohulupone, hulupone + adhulupone, and feruloyl quinic acid. Hop methanolic extracts have previously been shown to improve inflammation and reduce ear edema in mice [56]. Cohulupone was shown to increase natural killer cell activity against leukemia [57]. Hulupone is able to reduce monocyte and macrophage inflammation [58] and to act on the central nervous system to lessen the risk of depression [59]. Feruloyl quinic acid was previously described as an unstable compound with a strong anti-inflammatory capacity on macrophages [60]. The stability of our methanolic hop extract in targeting chondrocytes in vitro was unknown. To resolve this possible issue, we also vectorized the extract by using agro-based nanoliposomes from rapeseed lecithin. These systems have been previously employed to protect, carry, and release active molecules to chondrocytes [20]. Moreover, they are made of fatty acids (ω-3), which are known for their anti-inflammatory properties [24,25,26].

Physicochemical characterization revealed no difference between empty and hop-extract-loaded nanoliposomes in terms of size, size distribution, and colloidal stability. As previously described, cell culture conditions with high rapeseed lecithin concentrations (500 and 1000 µg/mL) in empty nanoliposomes or nanoliposomes encapsulating hop extract affected human chondrocyte viability in vitro [19,20]. To assess the effect of empty or loaded nanoliposomes on inflammation, the culture conditions were selected to be harmless to chondrocytes and to include the highest quantities of compounds (250 µg/mL lecithin with or without 75 µg/mL hop extract). When they were tested in an IL-1β-induced OA model of human chondrocytes in vitro, empty nanoliposomes seemed to have a greater anti-inflammatory effect than the hop extract alone in diminishing COX-2, iNOS, and mPGES-1 transcription. Nevertheless, this influence was switched to produce PGE_2_ and NO. Culture conditions with nanoliposomes encapsulating hop extract exhibited the most remarkable results in alleviating inflammation, whether for transcripts or end products. These results confirm that PGE_2_ [61,62,63] and NO [64,65,66] are important mediators of the inflammation involved in cartilage degeneration and that targeting their pathways is an efficient approach to preventing OA. Thus, we demonstrated that nanoliposomes are delivery systems that can carry anti-inflammatory compounds of natural origin and that our hop extract’s anti-inflammatory effect on human chondrocytes in vitro is potentiated by its encapsulation in rapeseed lecithin nanoliposomes. Nevertheless, the possibility that a part of the anti-IL-1β effect could be due to the antioxidative activity of hop-derived bitter acids cannot be excluded [33]. Consequently, it would be interesting to study this activity in another study using extracts from stems, leaves, cones, or a combination of the three parts (alone or encapsulated) [67,68]. In the future, it would also be interesting to evaluate the potential of other hop varieties extracted with the same method on chondrocyte inflammation or to test our system on other inflammation models. In addition, these results also show that actions can be implemented to move towards a “zero waste” chain in the cultivation of rapeseed and hops.

Although our results were obtained from human cells and are encouraging for the development of new therapeutics to prevent joint inflammation and improve OA, they were only achieved in vitro. These limitations have to be overcome to optimize treatment safety with further experiments to validate our system’s penetration in cartilage and its efficiency in various OA models. For example, hop-extract-loaded nanoliposomes could be appraised by an ex vivo approach with human cartilage explants or in vivo OA models. Among the various stated in vivo models, OA induction can be performed by administering intra-articular injections of IL-1β [69] or collagenase [70] in mice and mono-iodoacetate in rats [71]. It can also be triggered surgically in mice by destabilizing the medial meniscus [72] and can be induced spontaneously according to the rat genetic background [73]. The stability of the system could also be evaluated by testing the route (by mouth or by intra-articular injection) on small- and large-scale animal models. This could help to decipher nanoliposome interactions with tissues and adjust their efficacy in situ.

## 4. Materials and Methods

Reagents were obtained from Merck (Saint-Quentin-Fallavier, France), and media components were obtained from Lonza (Colmar, France), unless specified otherwise.

### 4.1. Plant Material

Hop rhizomes (*Humulus lupulus* L. cv. Cascade) were cultivated on potting soil in a greenhouse respecting organic farming conditions (no pesticides and no synthetic fertilizers). Six weeks after potting, fresh leaves were collected from young stems.

### 4.2. Extraction of Specialized Metabolites from Hop

Freshly collected leaves were frozen in liquid nitrogen and ground with a pestle and mortar. A double maceration extraction was performed on 1.1 g of frozen powder with 8 mL of MeOH 80% (MeOH/H_2_O 80/20 (*v*/*v*)) at room temperature. The sample was homogenized using an agitator for 10 min at 2000 rpm (Heidolph Instruments GmbH & Co. KG, Schwabach, Germany), sonicated at 37 kHz for 10 min in an ultrasonic bath (using “sweep” mode) (Elmasonic S 70, Singen, Germany), and finally centrifuged for 40 min at room temperature at 3234× *g*. Then, 7 mL of the supernatant was collected, and a second extraction was performed on the pellet in the same conditions. The extraction solution was dried in a speed vacuum device for 17 h (Eppendorf Concentrator Plus, Hamburg, Germany).

### 4.3. Characterization of Specialized Metabolites from Dry Extract

A total of 14.2 mg of dried extract was solubilized in 50 µL of MeOH 20% (MeOH/H_2_O 20/80 (*v*/*v*)) (agitation and sonication). The sample was centrifuged at 21,130× *g* for 10 min at 10 °C, and the supernatant was collected. Then, 38 µL of the extract was mixed with 1.9 µL of epoxyaurapten (15.9 mM), which was used as the internal standard.

### 4.4. UHPLC-ESI-MS Analysis

Chromatographic analyses were performed on a Vanquish UHPLC system equipped with a binary pump, an autosampler, and a temperature-controlled column. Metabolites contained in the extracts (10 µL) were separated on a ZORBAX Eclipse Plus C18 (95 Å, 100 × 2.1 mm, 1.8 µm; Agilent Technologies, Waldbronn, Germany) using a mobile-phase gradient (Table 1).

HRMS^1^ detection was performed on an Orbitrap IDX^TM^ (Thermo Fisher Scientific, Bremen, Germany) mass spectrometer in positive and negative electrospray ionization (ESI) modes. The capillary voltages were set at 3.5 kV and 2.35 Kv for positive and negative modes, respectively. The source gases were set (in arbitrary unit min^−1^) to 40 (sheath gas), 8 (auxiliary gas), and 1 (sweep gas), and the vaporizer temperature was 320 °C. Full-scan MS^1^ spectra were acquired from 120 to 1200 *m*/*z* at a resolution of 60,000. MS^2^ analysis was performed, too, using the AcquireX data acquisition workflow developed by Thermo Fisher. Briefly, this workflow increases the number of MS^2^ acquisitions, especially on low-intensity ions, through the creation of an inclusion list after the first injection of the sample and the establishment of a dynamic exclusion list occurring by an iterative sample analysis (involving 5–6 successive injections).

### 4.5. Molecular Identification

Compound Discoverer^TM^ software (version 3.3) was used to profile metabolites in our hop leaves extract.

### 4.6. Nanoliposome Preparation

Rapeseed lecithin was acquired from the Solae Europe SA society (Le Grand-Saconnex, Switzerland). Briefly, a total of 2.5 mL of distilled water was added to 50 mg of rapeseed lecithin and agitated under nitrogen. Samples were then probe-sonicated at 40 kHz for 8 min (1 s on, 1 s off) in an ice bath. The produced empty liposomes were stored in glass bottles protected from the dark at 4 °C until use. To prepare hop-extract-loaded nanoliposomes, 15 mg of hop extract was added to rapeseed lecithin, and then the mixture was prepared as explained above.

### 4.7. Physicochemical Characterization

#### 4.7.1. Dynamic Light Scattering

The average hydrodynamic particle diameter, particle-size distribution, polydispersity index, and ζ-potential for empty and hop-extract-loaded rapeseed lecithin liposomes were determined after the dilution of the samples in distilled water (1:200), followed by filtration (0.2 µm) using the DLS technique with a Zetasizer Nano ZS (Malvern Instruments Ltd., Worcestershire, UK).

#### 4.7.2. Transmission Electron Microscopy

To visualize liposome size and morphology, TEM was employed using a negative-staining method. Briefly, empty or hop-extract-loaded rapeseed lecithin liposomes were diluted in distilled water (1:10). A solution of the diluted samples and 2% phosphotungstic acid was placed for 5 min on a Formvar/carbon-supported copper grid (200 mesh, 3 mm diameter HF 36). The mesh was observed using a Philips CM20 operating at 200 kV, and micrographs were recorded using an Olympus TEM CCD camera.

### 4.8. Human Cartilage Specimens and Chondrocyte Culture

All specimen collections and all procedures were approved by the Ethics Committee of the Nancy University Hospital (agreement #UF 9607-CPRC 2005) and conducted in conformity with the declaration of Helsinki. Written informed consent was obtained from all participants.

Femoral condyles and tibial plateaus were obtained from OA patients (aged 65 ± 7 years, mean ± standard deviation (SD)) undergoing total knee replacement surgery according to the recognized clinical criteria of the American College of Rheumatology [74]. Cartilage sections were graded histologically according to Mankin’s score based on surface fibrillation, matrix depletion, cellularity, and the integrity of the tidemark [75].

Chondrocytes were isolated from femoral head caps. Cells were obtained by sequential digestion with pronase and collagenase, then washed twice in phosphate-buffered saline (PBS), and cultured to confluence in 75-cm^2^ flasks at 37 °C in a humidified atmosphere containing 5% CO_2_, as previously described [76]. Chondrocytes that recovered from the digestion were seeded at a density of 25,000 cells/cm^2^ in DMEM/Ham’s F-12 supplemented with L-glutamine (2 mM), penicillin (100 U/mL), streptomycin (100 µg/mL), and 10% heat-inactivated fetal calf serum (FCS; Thermo Fisher Scientific, Waltham, MA, USA). All experiments were performed using confluent cells at passage 1.

Stock solutions of hop extract (6 mg/mL), empty nanoliposomes (20 mg/mL of rapeseed lecithin), and nanoliposomes encapsulating hop extract (20 mg/mL of rapeseed lecithin and 6 mg/mL of hop extract) were prepared. Serial dilutions in the media were realized to obtain increasing concentrations of the hop extract alone (3 to 300 µg/mL), empty nanoliposomes (10 to 1000 µg/mL rapeseed lecithin), or nanoliposomes encapsulating hop extract (at 10 to 1000 µg/mL rapeseed lecithin and 3 to 300 µg/mL hop extract).

### 4.9. Biocompatibility Assays

To evaluate the impact of nanoliposomes on cell behavior, different parameters were estimated: cytotoxicity, cell metabolic activity, and cell proliferation.

#### 4.9.1. Cytotoxicity Assay

The cytotoxicity test was performed using the Cytotoxicity Detection Kit^PLUS^ (LDH) (#04744926001; Roche, France) according to the manufacturer’s instructions. The basis of this assay is the measurement of LDH activity released from damaged cell cytosol. Three controls are involved: high control (maximum LDH release), low control (untreated cells), and background control (assay medium). The absorbance was read on a spectrophotometer at 490 nm (Varioskan^®^ Flash, Thermo Scientific, Waltham, MA, USA). The average absorbance values of the triplicate samples and controls were calculated and subtracted from the background control absorbance values to determine the experimental absorbance values. The percentage of cytotoxicity was determined over the value of the high control (fixed to 100).

#### 4.9.2. Cell Metabolic Activity

Cell metabolic activity was measured using the MTT ((3-(4,5-dimethylthiazol-2-yl)-2,5-diphenyltetrazolium bromide) assay. First, 50 µL of MTT solution was added to 200 µL of cell culture medium. Briefly, chondrocytes were incubated for 4 h (5% CO_2_, 95% humidity at 37 °C) to allow the yellow dye to be transformed into blue formazan crystals by mitochondrial dehydrogenases. The supernatant was removed, and this insoluble product was protected from light and dissolved by adding 200 µL of dimethyl sulfoxide and gently mixing at 37 °C for 5 min. The supernatants were removed, protected from light, and centrifuged, and their absorbance was read within 30 min using a Varioskan^®^ Flash (Thermo Fisher Scientific, llkirch-Graffenstaden, France) at 540 nm. The control condition for chondrocyte metabolic activity was used as the reference value.

#### 4.9.3. Cell Proliferation

Cell proliferation was assessed after the culture of chondrocytes using the Hoechst assay, which allows cell DNA quantification. Briefly, chondrocytes were harvested from 12-well plates and suspended in 100 µL of Hoechst buffer (0.1 M of NaCl, 1 mM EDTA, and 10 mM TRIS, pH 7.4). This was followed by five series of freezing/thawing cycles to rupture cells and release their DNA. Black fat-bottom plates with low fluorescence background were used to perform the assay, which was quantified using a calf thymus DNA standard curve. Samples were mixed with 2 µL of Hoechst solution, and measurements of DNA samples and standards were performed by fluorescence spectrophotometry. Each sample’s DNA concentration was based on its fluorescence measurement compared to the standard curve, and the measurements of DNA samples and standards were performed by fluorescence spectrophotometry (360 nm excitation/460 nm emissions, Varioskan^®^ Flash, Thermo, France). The DNA concentration (µg/mL) of each sample was based on its fluorescence measurement relative to the standard curve.

### 4.10. RNA Isolation, Reverse Transcription, and Real-Time Polymerase Chain Reaction (RT-PCR)

Total RNAs were isolated from cultured chondrocytes using the Nucleospin RNA kit^®^ (Macherey Nagel, Hoerdt, Germany) according to the manufacturer’s instructions. A total of 200 nanograms of total RNAs was reverse transcribed at 37 °C for 90 min in a 20 μL reaction mixture containing 200 U Moloney Murine Leukemia Virus reverse transcriptase (Invitrogen, Fisher Scientific, Illkirch, France), 1.5 mM MgCl_2_, 5 μM random hexamer primers, and 10 mM dNTP. A Mastercycler gradient thermocycler (Eppendorf, Hamburg, Germany) was used to produce cDNAs. Subsequently, RT-PCR was completed using Step One Plus^TM^ (Applied Biosystems, Fisher Scientific, Illkirch, France) technology using specific primers (Table 2) and the iTAQ SYBRgreen^TM^ master mix system (Bio-Rad, Steenvoorde, The Netherlands). All reagents used for RT-PCR were added at the concentrations recommended by the manufacturer. The melting curve was generated to determine the melting temperatures of the specific PCR products, and after amplification, the product size was checked on a 1% agarose gel stained with Gel Red (Biotium, Interchim, Montlucon Cedex, France). Each run included positive and negative reaction controls. The mRNA levels of the gene of interest and of ribosomal protein 29 (RP29), chosen as a housekeeping gene, were determined in parallel for each sample. Quantification was determined using the ΔΔCt method, and the results were expressed as fold expression over the appropriate control.

### 4.11. PGE_2_ Assay

The levels of PGE_2_ were determined in culture supernatants using the Assay Design^®^ ELISA kit (Oxford Biomedical Research, Ann Arbor, MI, USA) according to the manufacturer’s instructions. The assay is based on the combined use of a monoclonal antibody against PGE2 and an alkaline phosphatase-conjugated polyclonal antibody. After the addition of paranitrophenylphosphate substrate, absorbance was read at 405 nm on a microplate reader (Varioskan^®^ Flash, Thermo Scientific, France). The limit of detection was 10 pg/mL, and the assay showed only little cross-reactivity with PGE_1_ (manufacturer’s data). Positive controls were used in each experiment.

### 4.12. Nitrite Assay

NO production was estimated spectrophotometrically by measuring the accumulation of nitrites in culture supernatants by the Griess reaction (Green LC et al., Anal Biochem 1982; 126:131-8). Briefly, 100 µL of culture supernatant was mixed with 100 µL of Griess reagent (1% of sulfanilamide in 2.5% H_3_PO_4_ and 0.1% of N-Naphtylethylenediamine dihydrochloride in H2O, *v*/*v*) for 5 min at room temperature in microtiter plates. The absorbance was measured at 550 nm with a microplate reader (Varioskan^®^ Flash, Thermo Scientific, llkirch-Graffenstaden, France), and the nitrite concentration was calculated with a standard curve of sodium nitrite ranging from 0 to 50 µM. The limit of quantification of this method was determined to be 1 µM nitrites.

### 4.13. Statistical Analyses

Results are expressed as the mean ± standard deviation. Statistical analyses were performed with GraphPad Prism 6 (GraphPad Software, San Diego, CA, USA) using one-way ANOVA multiple comparisons followed by Tukey correction. The *p* values are indicated in the legends if considered significant.

## Figures and Tables

**Figure 1 ijms-23-12423-f001:**
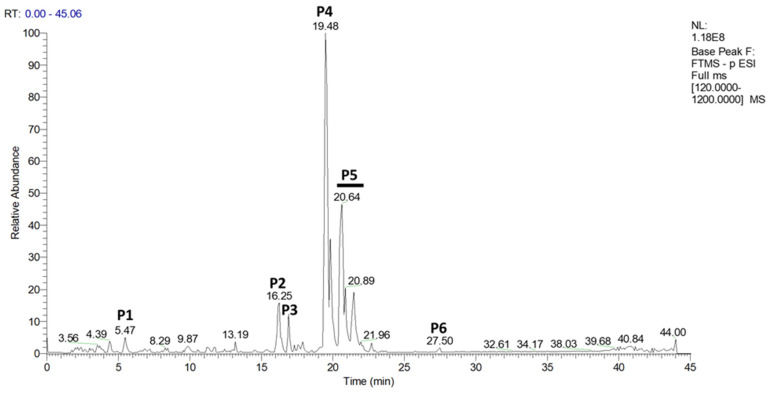
UHPLCMS chromatogram obtained from methanolic hop-leaf extract in negative mode. The molecular masses of the different products were characterized in negative mode. Their putative identifications were made using the Compound Discoverer^TM^ software (version 3.3). P1: *m*/*z* 367.1043 [M-H]^−^, feruloyl quinic acid; P2: *m*/*z* 327.2188 [M-H]^−^, not identified; P3: *m*/*z* 227.1298 [M-H]^−^, not identified; P4: *m*/*z* 317.1766 [M-H]^−^, cohulupone; P5: *m*/*z* 331.1926 [M-H]^−^, hulupone + adhulupone; P6: *m*/*z* 449.2555 [M-H]^−^, not identified. [M-H]^−^—negative ion chemical ionization mass spectra; *m*/*z*—mass-to-charge ratio; P—peak; RT—retention time; UHPLC—ultra-high-performance liquid chromatography; NL—normalization level; 1.18.10^8^—intensity of the most abundant ion; Base Peak—the most abundant ion is given at an intensity of 100 (relative abundance); FTMS—Fourier transform ion cyclotron resonance mass spectrometer; - p ESI—negative electrospray ionization; [120.0000–1200.0000]—*m*/*z* range measured. Green dashes show retention time of some peaks.

**Figure 2 ijms-23-12423-f002:**
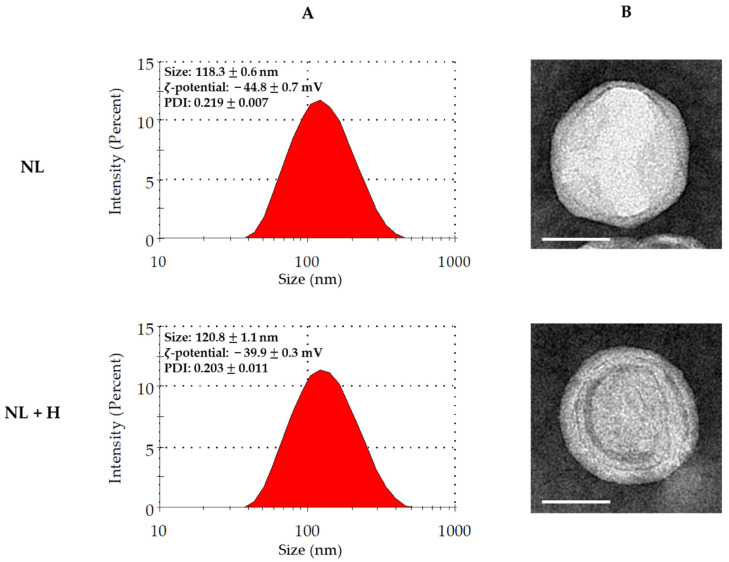
Physicochemical characterization of empty and hop-extract-loaded rapeseed lecithin liposomes. (**A**) Graphical representation of size distribution, average size, ζ-potential, and PDI of rapeseed lecithin nanoliposomes measured by DLS. (**B**) Representative TEM pictures of rapeseed lecithin liposomes. The white scale bar corresponds to 50 nm. DLS—dynamic light scattering; NL—empty nanoliposomes; NL + H—hop-extract-loaded nanoliposomes; PDI—polydispersity index; TEM—transmission electron microscopy; SD—standard deviation. The numerical data are represented as mean ± SD of at least three individual experiments.

**Figure 3 ijms-23-12423-f003:**
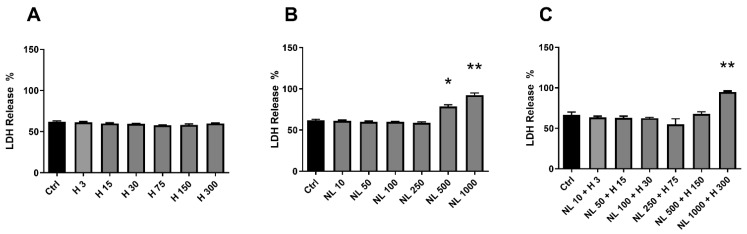
Evaluation of cytotoxicity with increasing concentrations of hop extract, empty nanoliposomes, and nanoliposomes encapsulating hop extract in chondrocytes. Human chondrocytes were exposed in vitro to increasing concentrations of (**A**) hop extract alone (3 to 300 µg/mL), (**B**) empty nanoliposomes (10 to 1000 µg/mL rapeseed lecithin), or (**C**) nanoliposomes encapsulating hop extract (at 10 to 1000 µg/mL rapeseed lecithin and 3 to 300 µg/mL hop extract). The potential cytotoxicity of each culture condition was measured by LDH assay on day 7. Ctrl—control cells, meaning chondrocytes not treated with hop extract alone, empty nanoliposomes, or nanoliposomes encapsulating hop extract; LDH—lactate dehydrogenase; H—hop extract alone; H 3—hop extract alone diluted to a final concentration of 3 µg/mL; NL—empty nanoliposomes; NL 10—empty nanoliposomes diluted to a final lecithin concentration of 10 µg/mL; NL + H—hop-extract-loaded nanoliposomes; NL 10 + H 3—nanoliposomes encapsulating hop extract diluted to a final lecithin concentration of 10 µg/mL and a final hop concentration of 3 µg/mL; SD—standard deviation. The reported data are represented as mean ± SD of at least four individual experiments. Significance compared to Ctrl is indicated as * for *p* < 0.01 and ** for *p* < 0.001.

**Figure 4 ijms-23-12423-f004:**
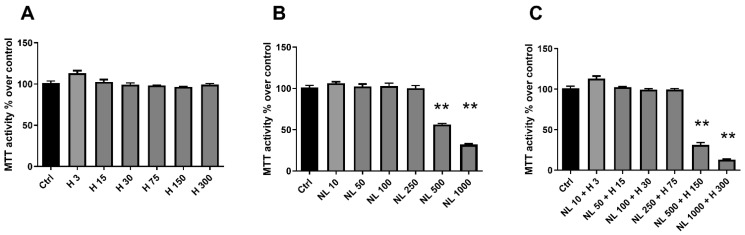
Evaluation of cell metabolic activity with increasing concentrations of hop extract, empty nanoliposomes, and nanoliposomes encapsulating hop extract in chondrocytes. Human chondrocytes were exposed in vitro to increasing concentrations of (**A**) hop extract alone (3 to 300 µg/mL), (**B**) empty nanoliposomes (10 to 1000 µg/mL rapeseed lecithin), or (**C**) nanoliposomes encapsulating hop extract (respectively at 10 to 1000 µg/mL rapeseed lecithin and 3 to 300 µg/mL). Cell metabolic activity for each culture condition was measured by MTT assay on day 7, where Ctrl was chosen as the reference value corresponding to an activity of 100%. Ctrl—control cells, meaning chondrocytes not treated with hop extract alone, empty nanoliposomes, or nanoliposomes encapsulating hop extract; H—hop extract alone; H 3—hop extract alone diluted to a final concentration of 3 µg/mL; MTT—3-(4,5-dimethylthiazol-2-yl)-2,5-diphenyltetrazolium bromide; NL—empty nanoliposomes; NL 10—empty nanoliposomes diluted to a final lecithin concentration of 10 µg/mL; NL + H—hop-extract-loaded nanoliposomes; NL 10 + H 3—nanoliposomes encapsulating hop extract diluted to a final lecithin concentration of 10 µg/mL and a final hop concentration of 3 µg/mL; SD—standard deviation. Significance compared to Ctrl is indicated as ** for *p* < 0.001.

**Figure 5 ijms-23-12423-f005:**
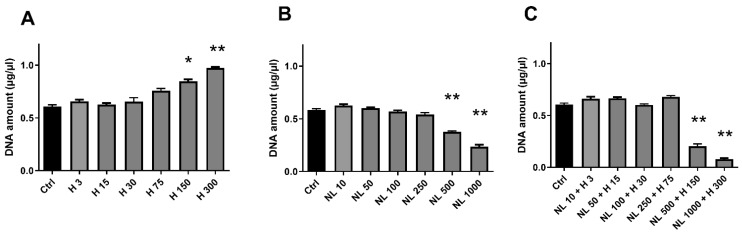
Evaluation of cell proliferation with increasing concentrations of hop extract, empty nanoliposomes, and nanoliposomes encapsulating hop extract in chondrocytes. Human chondrocytes were exposed in vitro to increasing concentrations of (**A**) hop extract alone (3 to 300 µg/mL), (**B**) empty nanoliposomes (10 to 1000 µg/mL rapeseed lecithin), or (**C**) nanoliposomes encapsulating hop extract (respectively at 10 to 1000 µg/mL rapeseed lecithin and 3 to 300 µg/mL). Cell proliferation in each culture condition was assessed by measuring DNA concentrations on day 7. Ctrl—control cells, meaning chondrocytes not treated with hop extract alone, empty nanoliposomes, or nanoliposomes encapsulating hop extract; H—hop extract alone; H 3—hop extract alone diluted to a final concentration of 3 µg/mL; NL 10—empty nanoliposomes diluted to a final lecithin concentration of 10 µg/mL; NL + H—hop-extract-loaded nanoliposomes; NL 10 + H 3—nanoliposomes encapsulating hop extract diluted to a final lecithin concentration of 10 µg/mL and a final hop concentration of 3 µg/mL; SD—standard deviation. The reported data are represented as mean ± SD of at least four individual experiments. Significance compared to Ctrl is indicated as * for *p* < 0.01 and ** for *p* < 0.001.

**Figure 6 ijms-23-12423-f006:**
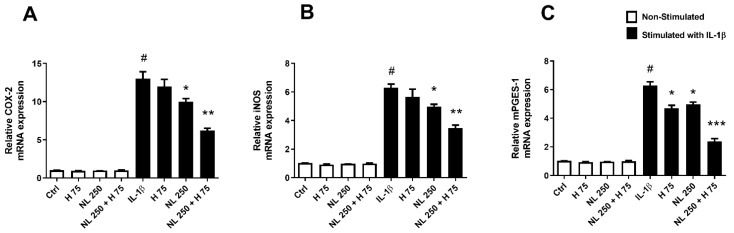
Influence of hop extract, empty nanoliposomes, and nanoliposomes encapsulating hop extract on inflammation marker transcription in vitro. Human chondrocytes were stimulated or not with 1 ng/mL IL-1β and exposed to hop extract alone (75 µg/mL), empty nanoliposomes (250 µg/mL rapeseed lecithin), or nanoliposomes encapsulating hop extract (respectively at 250 µg/mL rapeseed lecithin and 75 µg/mL). The culture conditions were performed for 6 h for early inflammation markers (COX-2 and iNOS mRNA) and 24 h (mPGES-1 mRNA). Total RNA was extracted, then reverse transcribed into cDNA, and finally analyzed by real-time PCR. The relative abundance of transcripts of (**A**) COX-2, (**B**) mPGES-1, and (**C**) iNOS was normalized to RP29. Quantifications were determined by using the ΔΔCt method. COX-2—cyclooxygenase-2; Ctrl—control cells, meaning chondrocytes not treated with hop extract alone, empty nanoliposomes, or nanoliposomes encapsulating hop extract; H—hop extract alone; H 75—hop extract alone diluted to a final concentration of 75 µg/mL; IL-1β—interleukin-1β; iNOS—inducible nitric oxide synthase; mPGES-1—microsomal prostaglandin E2 synthase-1; NL—empty nanoliposomes; NL 250—empty nanoliposomes diluted to a final lecithin concentration of 250 µg/mL; NL + H—hop-extract-loaded nanoliposomes; NL 250 + H 75—nanoliposomes encapsulating hop extract diluted to a final lecithin concentration of 250 µg/mL and a final hop concentration of 75 µg/mL; PCR—polymerase chain reaction; RP29—ribosomal protein 29; SD—standard deviation. The reported data are represented as mean ± SD of at least four individual experiments. Significance compared to Ctrl is indicated as # for *p* < 0.01. Significance compared to cells stimulated by IL-1β alone is indicated as * for *p* < 0.01, ** for *p* < 0.001, and *** for *p* < 0.0001.

**Figure 7 ijms-23-12423-f007:**
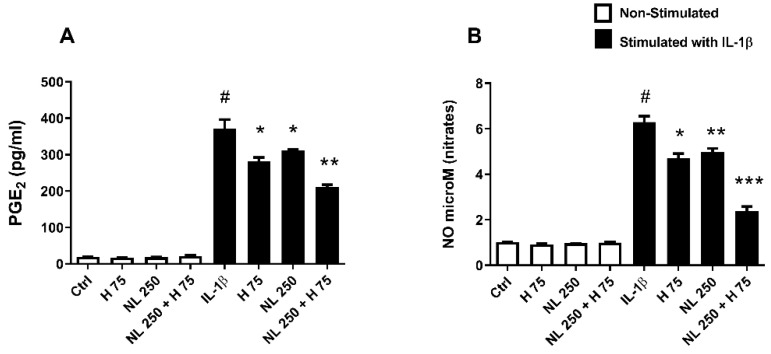
Influence of hop extract, empty nanoliposomes, and nanoliposomes encapsulating hop extract on inflammation marker release in vitro. Human chondrocytes were stimulated or not with 1 ng/mL IL-1β and exposed to hop extract alone (75 µg/mL), empty nanoliposomes (250 µg/mL rapeseed lecithin), or nanoliposomes encapsulating hop extract (at 250 µg/mL rapeseed lecithin and 75 µg/mL hop extract). The culture conditions were performed for 48 h. The release of (**A**) PGE_2_ was quantified by ELISA, and the production of (**B**) NO was assessed by the Griess reaction. Ctrl—control cells, meaning chondrocytes not treated with hop extract alone, empty nanoliposomes, or nanoliposomes encapsulating hop extract; ELISA—enzyme-linked immunosorbent assay; H—hop extract alone; H 75—hop extract alone diluted to a final concentration of 75 µg/mL; IL-1β—interleukin-1β; NL—empty nanoliposomes; NL 250—empty nanoliposomes diluted to a final lecithin concentration of 250 µg/mL; NL + H—hop-extract-loaded nanoliposomes; NL 250 + H 75—nanoliposomes encapsulating hop extract diluted to a final lecithin concentration of 250 µg/mL and a final hop concentration of 75 µg/mL; NO—nitric oxide; PGE_2_—prostaglandin E_2_; SD—standard deviation. The reported data are represented as mean ± SD of at least four individual experiments. Significance compared to Ctrl is indicated as # for *p* < 0.01. Significance compared to cells stimulated by IL-1β alone is indicated as * for *p* < 0.01, ** for *p* < 0.001, and *** for *p* < 0.0001.

**Table 1 ijms-23-12423-t001:** UHPLC-MS mobile-phase gradient elution conditions.

Time (min)	A (%)	B (%)
0	90	10
2	90	10
4	80	20
5	75	25
6.9	69	31
9	69	31
15	50	50
17	38	62
25	22.5	77.5
32	20	80
35	17	83
35.5	0	100
39	0	100
40	90	10
45	90	10

Mobile phase A—H_2_O + 0.1% formic acid (*v*/*v*); mobile phase B—acetonitrile + 0.1% formic acid (*v*/*v*).

**Table 2 ijms-23-12423-t002:** Sequences of specific primers for RT-PCR analyses.

Genes	Sequences 5′-3′
COX-2	Fwd: GCT-GGA-ACA-TGG-AAT-TAC-CCA
	Rev: CTT-TCT-GTA-CTG-CGG-GTG-GAA
mPGES-1	Fwd: TGG-TCA-TCA-AGA-TGT-ACG-TGG-T
	Rev: GGG-TCG-CTC-CTG-CAA-TAC-T
iNOS	Fwd: TGC-AAT-GAA-TGG-GGA-AAA-AG
	Rev: ATT-CTG-CTG-CTT-GCT-GAG-GT
RP29	Fwd: CTC-TAA-CCG-CCA-CGG-TCT-GA
	Rev: ACT-AGC-ATG-ATT-GGT-ATC-AC

COX-2—cyclooxygenase-2; mPGES-1—microsomal prostaglandin E_2_ synthase-1; iNOS—inducible nitric oxide synthase; Fwd—forward primer; RT-PCR—real-time polymerase chain reaction; Rev—reverse primer; RP—ribosomal protein.

## Data Availability

Not applicable.

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
