# Peer review of "Hop Extract Anti-Inflammatory Effect on Human Chondrocytes Is Potentiated When Encapsulated in Rapeseed Lecithin Nanoliposomes"

_ijms, 2022, doi:10.3390/ijms232012423_

Round 1

Reviewer 1 Report

The authors use nanoliposomes to formulated a batch of hop extract valorized from leaves to test their biocompatibility with increasing concentrations in human chondrocytes in vitro.

The manuscript is well written but the presentation of the results is not very clear. In the figures captions the authors have to add the statistic analysis and the significativity (p value). Furthermore, the manuscript laks the study of internalization of nanoliposomes in human chondrocytes cells. Confocal or TEM microscopy investigation could be useful to study the nanoliposomes localization.  

Author Response

Comments and Suggestions for Authors

The authors use nanoliposomes to formulated a batch of hop extract valorized from leaves to test their biocompatibility with increasing concentrations in human chondrocytes in vitro.

The manuscript is well written but the presentation of the results is not very clear. In the figures captions the authors have to add the statistic analysis and the significativity (p value).

The statistical analysis and the significance (p value) was already in the caption figure but in the middle. In order to facilitate the lecture to obtain this information, the caption figures were reshaped by adding the statistic information at the end of each.

Furthermore, the manuscript laks the study of internalization of nanoliposomes in human chondrocytes cells. Confocal or TEM microscopy investigation could be useful to study the nanoliposomes localization.

The investigation of the localization of nanoliposome in cell has been done on neuronal cells using confocal microscopy. This work is already accepted in Pharmaceutics (in Proof step). In this study, we showed that nanoliposome diffuse in the cells and after 48 hours nanoliposomes could be observed in the nucleus.

Reviewer 2 Report

I reviewed the manuscript "Hop Extract Anti-Inflammatory Effect on Human Chondrocytes is Potentiated When Encapsulated in Rapeseed Nanoliposomes". I find the work interesting and the results versatile for various diseases. I have some concerns regarding the liposomes.

1. First, I would specify "Rapeseed lecithin nanoliposomes" in the title and throughout the ms. Several plants are known to produce nanoparticles, missing this specification might be misleading for a not-so-careful or non expert reader.

2. In the methods, authors claim that they produce liposomes with 50 mg of lecithin, eventually with 12 mg of extract, in 2.5 mL water.

- Were liposomes purified after formulation? Please specify.

- Was encapsulation efficiency measured after formulation?

- The amount of hop extract used to label the formulations in the cell experiments was referred to the total amount of extract added? Please specify.

- Before DLS analysis samples were diluted (correct) and filtered: filtration hampers representativity of the analysis, as it removes all bigger liposomes and aggregates from the sample, no wonder that the PDI is small and the sample is homogeneous. Although I believe that results and conclusions are not heavily affected by this issue, this is an important flaw in the experimental design - unless authors used filtered nanoliposomes for the experiments, which would bring to a major list of missing controls. Authors should revise the dimensional analysis of nanoliposomes and provide more representative data without any filtration. 

- For the same reason, I would suggest authors to add TEM images where more than one particle is visible, to actually appreciate the size distribution of nanoliposomes.

- For cell experiments, authors say that the stock solutions of hop extract loaded nanoliposomes was at 20 mg/mL liposomes +12 mg/mL extract, but this is in contrast with the concentrations stated in section 4.6 (12 mg hop extract in 2.5 mL suspension). Am I missing something? Please explain.  

Overall I think the ms is well written, the English is clear and easy to follow. Please make sure that all decimal numbers have the right symbol (dots instead of commas, 2.5 instead of 2,5).

Author Response

Reviewer 2

Comments and Suggestions for Authors

I reviewed the manuscript "Hop Extract Anti-Inflammatory Effect on Human Chondrocytes is Potentiated When Encapsulated in Rapeseed Nanoliposomes". I find the work interesting and the results versatile for various diseases. I have some concerns regarding the liposomes.

  1. First, I would specify "Rapeseed lecithin nanoliposomes" in the title and throughout the ms. Several plants are known to produce nanoparticles, missing this specification might be misleading for a not-so-careful or non expert reader.

The manuscript was revised according to the reviewer’s comment.

  1. In the methods, authors claim that they produce liposomes with 50 mg of lecithin, eventually with 12 mg of extract, in 2.5 mL water.

- Were liposomes purified after formulation? Please specify.

The authors don’t really catch what the reviewer mean by purification “after formulation”, but after sonication the formulation wasn’t changed unless a filtration to insure the sterility of the solutions.

- Was encapsulation efficiency measured after formulation?

Encapsulation efficiency has been measured reaching more than 99% of encapsulation of hop extract.

- The amount of hop extract used to label the formulations in the cell experiments was referred to the total amount of extract added? Please specify.

Indeed, as the encapsulation efficiency was quite total, the label formulation at cell contact was referred to the total amount of extract added in the formulation. The concentration was tuned by different dilutions of the formulation in the medium.

- Before DLS analysis samples were diluted (correct) and filtered: filtration hampers representativity of the analysis, as it removes all bigger liposomes and aggregates from the sample, no wonder that the PDI is small and the sample is homogeneous. Although I believe that results and conclusions are not heavily affected by this issue, this is an important flaw in the experimental design - unless authors used filtered nanoliposomes for the experiments, which would bring to a major list of missing controls. Authors should revise the dimensional analysis of nanoliposomes and provide more representative data without any filtration. 

In previous studies, we have checked the impact of filtration by comparing before and after filtration and we didn’t observe significant differences toward size and nanoliposome concentration.

- For the same reason, I would suggest authors to add TEM images where more than one particle is visible, to actually appreciate the size distribution of nanoliposomes.

TEM images were done to see the shape and the lamellarity of the nanoliposomes. Unfortunately, measuring the size with this technique is useless as the liposome must be treated by changing its environment which has a great impact on size. Moreover, to have a global idea of the population size of nanoliposome, a very high number of particles must be treated on several images. Considering all these points, TEM is not an adequate tool to measure particle size.

- For cell experiments, authors say that the stock solutions of hop extract loaded nanoliposomes was at 20 mg/mL liposomes +12 mg/mL extract, but this is in contrast with the concentrations stated in section 4.6 (12 mg hop extract in 2.5 mL suspension). Am I missing something? Please explain.

The authors deeply thank the reviewer #2 for his/her careful reading and this comment. An error was made while transcribing information from a laboratory notebook. 5 was mistaken for 2. The stock solution of hop extract loaded nanoliposomes was at 20 mg/mL liposomes (50 mg of rapeseed lecithin in 2.5 mL suspension) + 6 mg/mL of hop extract (15 mg of extract in 2.5 mL suspension). Anyway, the hop extract concentrations presented in the manuscript suffered miscalculation even if they had the same order of magnitude than the right concentrations. The right concentrations have been recalculated and have been added to the last version of the manuscript.

A serial dilution of the stock solutions was made (stock solutions: NL 20,000 µg/mL and/or 6000 µg/mL).

Dilution

1:20

1:40

1:80

1:200

1:400

1:2000

Lecithin concentration (NL µg/mL)

1000

500

250

100

50

10

Hop extract concentration (H µg/mL)

300

150

75

30

15

3

Overall I think the ms is well written, the English is clear and easy to follow. Please make sure that all decimal numbers have the right symbol (dots instead of commas, 2.5 instead of 2,5).

The manuscript was edited according to the reviewer’s comment.

Round 2

Reviewer 2 Report

I appreciate the authors' work to improve their manuscript. 

Purification is a step that is needed after the formulation of nanosystems to separate the formed particles from the free drug that was not incorporated. For liposomes, dyalisis or size exclusion chromatography are widely used to purify free drug from loaded particles, and after EE% can be measured by quantifying either the free drug or the encapsulated part after a solvent extraction. I still think authors missed this part and they should add this step in their future work, however I don't think this affects the results.

Overall, I recommend for publication.